# Evaluating the Implementation of a Multicomponent Intervention Consisting of Education and Feedback on Reducing Benzodiazepine Prescriptions by General Practitioners: BENZORED Hybrid Type I Cluster Randomized Controlled Trial

**DOI:** 10.3390/ijerph18157964

**Published:** 2021-07-28

**Authors:** Isabel Socias, Alfonso Leiva, Haizea Pombo-Ramos, Ferran Bejarano, Ermengol Sempere-Verdú, Raquel María Rodríguez-Rincón, Francisca Fiol, Marta Mengual, Asunción Ajenjo-Navarro, Fernando Do Pazo, Catalina Mateu, Silvia Folch, Santiago Alegret, Jose Maria Coll, María Martín-Rabadán, Caterina Vicens

**Affiliations:** 1Healthcare Centre Manacor, Balearic Health Service IbSalut, 07500 Manacor, Spain; imsocias@ibsalut.caib.es; 2Balearic Islands Health Research Institute (IdISBa), 07120 Palma, Spain; 3Reseach Unit Mallorca, Balearic Health Service IbSalut, 07003 Palma, Spain; 4Primary Care Research Unit of Biscaia, Basque HealthCare Service Osakidetza, BioCruces Health Research Institute, 48903 Bizkaia, Spain; Haizea.pombo@osakidetza.eus; 5Catalunya Health Services-CatSalut, DAP Camp de Tarragona, 43002 Tarragona, Spain; fbejarano.tgn.ics@gencat.cat (F.B.); mmengual.tgn.ics@gencat.cat (M.M.); sfolch.tgn.ics@gencat.cat (S.F.); 6Paterna Healthcare Centre, Conselleria de Sanitat Universal i Salut Pública, 46980 Valencia, Spain; meresempere@gmail.com (E.S.-V.); ajenjo1903@hotmail.com (A.A.-N.); 7Pharmacy Department, Hospital Universitari Son Espases, Balearic Health Service IbSalut, 07120 Palma, Spain; raquel.rodriguez@ssib.es (R.M.R.-R.); fernando.dopazo@ssib.es (F.D.P.); 8Son Serra-La Vileta Healthcare Centre, Balearic Health Service IbSalut, 07013 Palma, Spain; francisca.fiolgelabert@ibsalut.es (F.F.); cmateus@telefonica.net (C.M.); santiagoalegret@hotmail.com (S.A.); caterinavicens@gmail.com (C.V.); 9Menorca Primary Care Management, Balearic Health Service IbSalut, 07701 Maó, Spain; josemaria.coll@ssib.es; 10Can Misses Healthcare Centre Ibiza, Balearic Health Service IbSalut, 07800 Ibiza, Spain; mmartinrabadan@asef.es

**Keywords:** benzodiazepines, adverse effects, primary health care, deprescribing, clinical trial

## Abstract

Background: General practitioners (GPs) in developed countries widely prescribe benzodiazepines (BZDs) for their anxiolytic, hypnotic, and muscle-relaxant effects. Treatment duration, however, is rarely limited, and this results in a significant number of chronic users. Long-term BZD use is associated with cognitive impairment, falls with hip fractures, traffic accidents, and increased mortality. The BENZORED IV trial was a hybrid type-1 trial conducted to evaluate the effectiveness and implementation of an intervention to reduce BZD prescription in primary care. The purpose of this qualitative study was to analyze the facilitators and barriers regarding the implementation of the intervention in primary care settings. Methods: A qualitative interview study with 40 GPs from three Spanish health districts. Focus group meetings with GPs from the intervention arm of the BENZORED IV trial were held at primary healthcare centers in the three districts. For sampling purposes, the GPs were classified as high or low implementers according to the success of the intervention measured at 12 months. The Consolidated Framework for Implementation Research (CFIR) was used to conduct the meetings and to code, rate, and analyze the data. Results: Three of the 41 CFIR constructs strongly distinguished between high and low implementers: the complexity of the intervention, the individual Stage of Change, and the key stakeholder’s engagement. Seven constructs weakly discriminated between the two groups: adaptability in the intervention, external policy and incentives, implementation climate, relative priority, self-efficacy, compatibility, and engaging a formally appointed implementation leader. Fourteen constructs did not discriminate between the two groups, six had insufficient data for evaluation, and eleven had no data for evaluation. Conclusions: We identified constructs that could explain differences in the efficacy in implementation of the intervention. This information is relevant for the design of successful strategies for implementation of the intervention.

## 1. Introduction

General practitioners (GPs) commonly prescribe benzodiazepines (BZDs) for their anxiolytic, hypnotic, and muscle relaxant effects, but often duration of treatment is longer than clinical indication, with a progressive increase in prescriptions and long-term consumers of BZD.

There is high variability in the prescription of BZDs among different countries. In the last decade many European countries have reduced the rate of BZD prescriptions, but in Spain, it is still high. Spain has an average consumption of 32.5 daily doses of hypnotics and sedatives (ATC: N05C) per 1000 inhabitants and 56.8 daily doses of anxiolytics (ATC: N05B) per 1000 inhabitants [1].

Long-term use of BZD can lead to dependence, tolerance, and an increased risk of falls and hip fracture [2,3,4]. It may be related with cognitive impairment [5,6,7,8] and increase the risk for traffic accidents [9,10,11] and mortality [12,13,14]. BZD misuse is a growing public health problem. Among BZD users, recent research indicated that 17.1% misused BZDs and 1.5% had BZD use disorders [15].

Due to physical dependence and tolerance, long-term users experience withdrawal symptoms and the worsening of anxiety or insomnia when they abruptly stop consumption, making it more difficult for patients to stop. GPs could prevent dependence limiting the duration of treatment when a BZD is first prescribed and should consider withdrawal for patients who are long-term users. Ribeiro et al. performed a systematic review of strategies for deprescribing BZDs and reported that interventions which aimed to educate patients and raise their awareness about dependence and adverse outcomes were more successful [16]. Evidence-based withdrawal interventions recommend a gradual dose reduction over time [17,18,19]. However, these strategies are poorly implemented in primary care.

The average time for an evidence-based practice recommendation to be incorporated into routine practice is 17 years [20], and only about half of all evidence-based practice recommendations are generally accepted in clinical practice [21]. More than 70% of an organization’s efforts to implement changes in clinical practice are not successful [22]. Evidence-based practice (EBP) can improve health outcomes for patients, but many factors can interfere with the adoption of an EBP [23]. Barriers identified to the deprescribing of BZD are the lack of institutional support to promote deprescription, the need of GPs for another treatment to offer, GPs’ lack of knowledge about effective interventions for withdrawal, and the need to overcome patient withdrawal symptoms. Moreover, patient perceptions of BZD tend to maximize the benefits and minimize the risks, making patients reluctant to stop treatment [16].

The BENZORED phase IV trial has a hybrid type-1 design to evaluate the effectiveness and the process of implementing an intervention based on a GP training workshop on the appropriate initial prescription of BZDs and on deprescribing BZD in long-term users, monthly feed-back about BZD prescriptions, and access to a support webpage. The study protocol was published previously [24]. In the present manuscript, we report the results of the implementation process, a qualitative evaluation of barriers, and facilitators to implement the intervention. Our study will shed light on factors that may influence the implementation of the intervention and will facilitate the development of an implementation strategy as an update of the interventions currently used in primary care settings.

## 2. Materials and Methods

### 2.1. Design

Qualitative interviews were conducted with 40 GPs form three Spanish health districts. The study protocol was approved by the Balearic Islands Ethical Committee of Clinical Research (IB3065/15), l’IDIAP Jordi Gol Ethical Committee of Clinical Research (PI 15/0148), and the Valencia Primary Care Ethical Committee of Clinical Research (P16/024). This trial was registered with the ISCRTN (ISRCTN28272199).

### 2.2. Setting and Participants

This qualitative study evaluated the implementation of the BENZORED IV intervention in primary health centers (PHCs) of 3 districts in Spain: Balearic Islands (Ib-Salut), Catalonia (Institut Català de la Salut; Tarragona-Reus district), and Valencia (Conselleria de Salut Universal; Arnau de Vilanova-Llíria district). We used the Consolidated Criteria for Reporting Qualitative Research (COREQ) checklist to guide the reporting of this study [25].

Focus group (FG) meetings were organized in a PHC in each district after the 12-month intervention. Participants were the GPs from the BENZORED IV trial allocated to the intervention arm. GPs were classified as high implementation GPs (HIGPs) if they prescribed significantly fewer and as low implementation GPs (LIGPs) if they prescribed the same amount or more BZDs after 12 months. We invited 4 GPs (2 HI and 2 LI) from each participating PHC. The participants were contacted by telephone.

One high implementation focus group (HIFG) and one low implementation focus group (LIFG) were organized in each Health District, but in the Valencia District, only one HIFG and two semi-structured telephone interviews were conducted with two LIGPs. All physicians signed consent for participation and agreed to be audio recorded and were offered a EUR 60 bonus as incentive.

### 2.3. Data Collection

The FG meetings were conducted between June and July 2018, each group had about 6 to 9 participants, and the meetings lasted 90 to 120 min. Telephone interviews lasted 30 to 40 min. Two researchers with experience in leading groups moderated the FGs, and several members of the research team participated as observers and took notes during the FG. All FG meetings and interviews were audio recorded, transcribed verbatim, and checked for accuracy. There was no need to repeat any telephone interview or FG meeting.

The Consolidated Framework Implementation Research (CFIR) was used to guide development of the FG meetings and for coding and data analysis [26,27,28]. The CFIR is a theoretical framework that provides a list of 41 constructs organized in five domains that can negatively or positively influence implementation:(1)Intervention Characteristics: Intervention Source, Evidence Strength and Quality, Relative Advantage, Adaptability, Trialability, Complexity, Design Quality and Packaging, Cost.(2)Outer Setting: Patient Needs and Resources, Cosmopolitanism, Peer Pressure, External Policies and Incentives.(3)Inner Setting: Structural Characteristics, Networks and Communications, Culture, Implementation Climate with 6 Sub-constructs, Readiness for Implementation with 3 Sub-constructs.(4)Individuals Characteristics: Knowledge and Beliefs about the Intervention, Self-efficacy, Individual Stage of Change, Individual Identification with Organization, Other Personal Attributes.(5)Process: Planning, Engaging with 6 Sub-constructs, Executing, Reflecting and Evaluating.

We identified the health districts as the “Outer Setting” domain and the PHCs as the “Inner Setting” domain.

The FG meetings began with open questions about each CFIR domain (Appendix A). This was followed by questions related to some specific CFIR constructs previously selected by the research team as keys to the implementation process, if these issues were not already addressed during the session of general questions.

### 2.4. Data Coding and Rating

Each transcript was coded and rated independently by two analysts. After carefully reading the transcripts, they collected all GPs statements for each construct and imported them into an Excel document for coding, rating, and analysis. A deductive approach for data coding was used to apply the CFIR codebook (available at www.wiki.cfirwiki.net, accessed on 2 June 2021). Analysts compared their codes in regular meetings, discussed the differences, and agreed on the final codes.

To rate the statements on each construct, the analysts followed the CFIR Rating Rules (available at cfirwiki.net) and used a qualitative consensus-based rating process. These rules rate the construct’s statements based on the valence (positive or negative influence) and strength (weak (1) or strong (2)). Thus, the range of rating was “−2” to “+2”, and “0” is a neutral rating. X indicated a mixed rating with equally positive and negative statements, but it could be with mainly positive statements (+*) or mainly negative statements (−*). All ratings were shared and agreed upon by the analysts.

### 2.5. Data Analysis and Interpretation

A table was developed with the ratings of each construct in each group, with the HIFGs in one column and the LIFGs in another column. This facilitated the identification of patterns in ratings of the CFIR constructs that distinguished these two groups.

Constructs were identified as:-Discriminatory if the construct weakly or strongly distinguished the HI and LIFGs;-Non-discriminatory if the construct did not distinguish the HI and LIFGs;-No data or insufficient data if the effect of the construct could not be assessed.

Two analysts independently analyzed and interpreted the constructs’ ratings. Decisions were shared, differences were discussed, and the final interpretation was agreed upon.

## 3. Results

A total of five FG meetings and two semi-structured individual telephone interviews took place, and these included 40 GPs (22 HIGPs and 18 LIGPs). Table 1 shows the characteristics of participating GPs.

Three of the 41 constructs strongly discriminated between the HI and LIFG (Intervention Complexity, Individual Stage of Change, and Engagement Key Stakeholders) and seven constructs weakly discriminated between these two groups (Intervention Adaptability, External Policy and Incentives, Implementation Climate, Compatibility, Relative Priority, Self-Efficacy, and Engaging a Formally Appointed Implementation Leader). Fourteen constructs were not discriminatory. Six constructs had insufficient data for evaluation and 11 constructs had no data for evaluation (Table 2).

We further described the discriminatory constructs in each CFIR domain using supportive statements from the participants (Table 3). We provided brief descriptions of some constructs that were not discriminatory but may be important to consider for a future implementation strategy. We also noted the occurrence of data saturation.

Domain I: Intervention Characteristics

Complexity was a strong discriminatory construct. The HIGPs perceived the intervention as having low complexity, well-conceived, having steps that were clear and concise, and not requiring great effort.

One LIFG perceived the intervention as having high complexity because asking a patient about BZD consumption meant “opening a Pandora’s box” that could lead to emotional stress and necessitate recommendations for psychotherapy sessions, delays in the consultation, and difficulties in involving the patient. The other two LIFG stated that they did not find the intervention very complex, but LIGPs reported that they were unclear about BZD’s discontinuation plan and that they often had to review the instructions.

Adaptability was a weakly discriminatory construct. The HI and LIFGs supported the adaptation of BZD withdrawal to patients by making it slower or faster, but the HIFGs provided more supportive, stronger, and more creative comments regarding the tailoring of BZD withdrawal for each patient.

All GPs perceived the intervention as an internal intervention (Intervention Source) and valued that those who designed the intervention were also GPs and placed greater value on the scientific evidence regarding its implementation (Evidence Strength and Quality).

In general, GPs stated that the intervention required more time than their usual clinical practice. Some comments indicated they believe the intervention offered advantages over their usual practice (Relative Advantage), but others believed the intervention greatly delayed consultation and offered no advantage. The training workshop and the support materials (Design Quality and Packaging) were valued as enablers of implementation, but some LIGPs found the workshop too theoretical and requested a workshop that provided more practical advice.

Domain II: Outer setting (Health District)

External Policy and Incentives was a weakly discriminatory construct. We distinguished health districts that did and did not have local policies regarding BZD prescribing including indicator and incentives.

GPs in the health district with local policies regarding BZD prescribing stated that knowing that the goals of the intervention were aligned with the goals of their health district encouraged them to implement the intervention. They perceived a greater willingness of patients to participate in the intervention because patients received the same information from multiple sources (brochures in the waiting rooms, comments between patients, dialogue with GPs). HIGPs made stronger supportive comments about the intervention and local policies on BZD and were more willing to adopt it. One LIFG without district health local policies made reference to the lack of global strategies aimed at professionals in all levels of care (hospital, primary care, nursing homes, etc.).

Domain III: Inner setting (Healthcare Center)

Implementation Climate was a weakly discriminatory construct. The HIFGs referred to the high degree of participation of their team in the project, to their high capacity to accept new challenges, and the role of the medical director as promoter. However, comments by the LIFGs were completely opposite.

Compatibility was a weakly discriminatory construct. Although all GPs believed that implementing the intervention required more time than is usually available per patient, HIGPs adapt their time and implemented it more when their workload was light. However, the LIGPs reported that the intervention was not compatible with workflows, because if they did, they caused significant delays.

Relative priority was a weakly discriminatory construct. All physicians stated that it was necessary to address the high rate of BZD prescriptions (Tension for Change), but only the HIGPs prioritized implementation of the intervention. Some HIGPs attached great importance to the project because it made them aware of the high consumption of BZDs and the need to address this problem.

In the Access to Knowledge and Information construct, we assessed the access to the support webpage, which has a video describing intervention instructions and supporting materials. GPs reported technical problems accessing the website. The GPs who were able to access the website rated the support materials as being high quality.

Domain IV: Individual Characteristics (GPs)

In the Self-efficacy construct, we were interested in determining the self-efficacy of the GPs for each component of the intervention and therefore divided this construct into two sub-constructs:First prescription: This was a non-discriminatory sub-construct. All GPs stated that correctly making the first BZD prescription by limiting the duration of treatment was the easiest part of the intervention and the most effective in preventing chronic BZD consumption.Benzodiazepine withdrawal: This was a strongly discriminatory sub-construct. Despite the difficulty of withdrawing from BZD treatment, HIGPs reported that the intervention offered them a useful tool to assist chronic users to withdraw from BZDs, and they saw clear benefits for patients. The LIGPs stated that BZD withdrawal seemed too difficult and in most cases was not even worth trying.

Individual Stage of Change was a strongly discriminatory construct. The HIGPs stated that they adopted the intervention and want to keep it in their practices. The LIGPs were enthusiastic about implementing the intervention soon after receiving the training, but they lost enthusiasm over time and began to prioritize other activities.

All GPs stated that the doctor-patient relationship and the patient’s trust in the GP were key points in being able to initiate BZD withdrawal (Knowledge and Beliefs about the Intervention).

Domain V: Process

Engaging a Formally Appointed Implementation Leader was a weakly discriminatory construct. Leaders were valued by HIFGs but by only one LIFG as key elements to drive implementation. The other two LIFGs made positive assessments of the leaders for their personal characteristics but not as drivers of implementation.

Stakeholders Engagement was a strongly discriminatory construct. The HIGPs were enthusiastic, involved in applying an innovation that offered a clear benefit to patients, incorporated it into their clinical practices, and strove to make it compatible with workflows. In contrast, the LIGPs were not so enthusiastic about the intervention.

All GPs mentioned the difficulty of getting patients to commit to BZD withdrawal (Engaging Innovation Participants) because they had been taking it for many years without noticing adverse effects and expressed fear of not being able to sleep if BZD was withdrawn. One HIFG made strong comments that patients were more aware of the need for BZD withdrawal because of the local policies regarding BZD prescription. These HIGPs were sometimes surprised by their success in engaging patients and achieving withdrawal or dose reductions.

All GPs agreed on the role of receiving individual feedback to promote implementation.

At the conclusion of the FG meeting, GPs were asked to suggest how the intervention could be improved to help develop a successful implementation strategy (Appendix A).

## 4. Discussion

We analyzed constructs related to the successful implementation of the intervention. Three constructs had strong discrimination: Complexity, Individual Stage of Change, and Engaging the key stakeholder. We also identified seven constructs as having weak discrimination: Adaptability, External policy and incentives, Implementation climate, Relative priority, Self-efficacy, Compatibility, and Engaging a formally appointed implementation leader.

The complexity was a strongly discriminatory construct. The LIGPs declared a high sense of complexity for the withdrawal of BZD in chronic users. LIGPs were unclear about BZD’s discontinuation plan and their heavy workload did not allow them to review the support materials with instructions as they needed. They also reported difficulty accessing the web. The training workshop was generally valued as a driver of implementation, but some LIGPs considered it impractical and too theoretical. However, HIGPs stated that the intervention offered them a useful tool to withdraw long-term users and wanted to maintain it in their practice, even though HIGPS also considered BZD deprescribing as difficult but not complex. Despite the perceived difficulty, they valued the intervention as flexible and integrable into GP practice [29].

All GPs reported that the first BZD prescription, which limits the duration of treatment, was the easiest component and where they perceived its highest self-efficacy.

Previous studies reported that managerial support can promote implementation of innovations [30,31]. In our study, some LIGPs requested local policies regarding BZD, and the GPs from the health districts with these policies corroborated that managerial support aligned with the intervention increases its visibility [31] and reinforced the importance of implementing it at both the patient and the physician level. As Ribeiro et al. suggested, educational interventions for patients are effective in increasing BZD discontinuation [16].

We found that the lack of time and the workload of GPs are important barriers to implementation in primary care [30,31,32,33]. The provision of more formal time clearly facilitated the implementation of an intervention [31]. HIGPs expressed that they were more likely implement the intervention when their workloads were light, so they could schedule longer initial appointments for chronic BZD users and avoid delays in consultations. Follow-up visits did not last longer than regular visits.

GPs declared a strong need to decrease prescriptions of BZDs but only the HIGPs prioritized the intervention and mentioned the influence of their medical director as a champion. Champions can play a positive role in improving implementation [26,29,34,35] by overcoming barriers and engaging their peers to adopt and maintain the intervention in regular practice [31]. HIGPs also valued the role of project leaders because of their accessibility and credibility. It should be noted, however, that HIGPs were older, more likely to be women, and had been working less time in the PHCs that participated in the study compared to LIGPs. The lack of an implementation leader could be a barrier to implementing an intervention in primary care [32]. GPs reported negative experiences when participating in projects in which there was not a project leader who could address problems.

Our findings suggest the importance of reducing the perceived complexity of the discontinuation of BZDs by chronic users to improve the GPs’ self-efficacy, engagement, prioritization, and adoption of the proposed intervention in clinical practice. We believe that reducing the perceived complexity in BZD discontinuation requires GPs to have extensive knowledge of the intervention. Notably, it is necessary to emphasize the importance of selecting the right patients for deprescribing BZDs in the training workshop for three major reasons. First, the patients should have sufficient emotional stability. Second, the patient must be receptive when receiving information about the harms of chronic BZD use and must understand the importance of gradually reducing the dose. Third, the patient must be willing to discuss his or her beliefs and fears regarding acceptance of BZD discontinuation. If these conditions are not met, BZD withdrawal will be difficult and be perceived as too complex. It is also important to resolve problems with web access in order to use the support materials.

The study results also suggest the role of champions in PHC to improve the implementation climate and stakeholder engagement. They could hold reminder meetings, as suggested by the GPs, to facilitate strong communication about the intervention and to make the team aware of the importance of prioritizing and adopting the intervention [27,29,36]. All GPs also valued monthly feedback to improve engagement. It has been reported that feedback is a key element to encourage implementation [29,37].

## 5. Strengths and Limitations

The main strength of the study is the systematic approach that we have carried out using the CFIR to evaluate the implementation process of the intervention before the health district promotes its implementation in primary care. The study was planned a priori to evaluate the CFIR domains impacting upon the implementation from the viewpoint of GPs, and we incorporate GPs’ proposals for improvement and evaluated the contextual factors. However, as the process evaluation was conducted at the completion of the trial, it is possible that participants may not have recalled some details of the implementation. It could be that a mid-process evaluation would have provided more information. Another limitation is that the study findings were not sent to the GPs for feedback; this feedback may have enriched our interpretation of the results. The analysts were not blinded to the rating and analysis of the data, so the potential for bias may exist. Patient satisfaction, especially with the BZD withdrawal intervention, was not assessed.

## 6. Conclusions

We identified several factors that influenced the implementation of an intervention that seeks to reduce prescriptions of BZDs in the primary care setting; these factors included the perceived complexity of the intervention and the role of local policies in promoting BZD deprescription. We also compiled suggestions for improvements that may help redefine the intervention. We believe that our results could be useful for organizations that decide to develop an implementation strategy to incorporate this intervention into the routine practice of GPs.

## Figures and Tables

**Table 1 ijerph-18-07964-t001:** Characteristics of participating GPs in the LIGPS and HIGPs categories.

	LIGPs (*n* = 18)	HIGPs (*n* = 22)
Age in years (mean, SD)	53 ± 6	57.3 ± 4.9
Females (n/N (%))	10/18 (55.5%)	13/22 (59.1%)
Years in the Health Care Center (mean, SD)	18 ± 9	16.4 ± 8.3

**Table 2 ijerph-18-07964-t002:** Ratings assigned to CFIR construct by group.

		Low Implementation Focus Groups			High Implementation Focus Groups			
		FG1	FG4	FG6	FG2	FG3	FG5	
**I. INTERVENTION CHARACTERISTICS**								
A	Intervention Source	+2	+1	+1	ND	+1	+2	
B	Evidence Strength and Quality	+2	0	+1	+1	+1	+1	
C	Relative Advantage	−1*	+1	ND	ND	ND	+1	
D	Adaptability	+1	+1	+1	+2	+1	+2	*
E	Trialability	ND	ND	ND	ND	ND	ND	
F	Complexity	−1	+1	+1	+2	+2	+1	**
G	Design Quality amd Packaging	+1	X	X	+2	+1	+1	
H	Cost	ND	ND	ND	ND	ND	ND	
**II. OUTER SETTING**								
A	Patient Needs & Resources	+1	+1*	ND	+1	ND	+1	
B	Cosmopolitanism	ND	ND	ND	ND	ND	ND	
C	Peer Pressure	+1	+1*	+1*	ND	+1*	+1	
D	External Policy & Incentives	−1*	+1	−1*	+1*	0	+2	*
**III. INNER SETTING**								
A	Structural Characteristics	−1	ND	ND	ND	ND	ND	
B	Networks and Communications	−1*	ND	ND	−1	ND	ND	
C	Culture	X	−1*	−1	ND	ND	ND	
D	Implementation Climate	−1	ND	−1	+2	+1	+1	*
1	Tension for Change	+1	+1	+1	+2	+1	+1	
2	Compatibility	−2	+1*	−2	−1	+1*	+1*	*
3	Relative Priority	−1*	−1	−1	+1*	+2	ND	*
4	Organizational Incentives and Rewards	ND	ND	ND	ND	ND	ND	
5	Goals & Feedback	ND	ND	ND	ND	ND	ND	
6	Learning Climate	ND	+1	+1	ND	+2	X	
E	Readiness for Implementation	ND	ND	ND	ND	ND	ND	
1	Leadership Engagement	ND	ND	ND	ND	ND	ND	
2	Available Resources	ND	ND	ND	−1	ND	ND	
3	Access to Knowledge and Information	ND	−2	+1*	ND	−1	−1	
**IV. CHARACTERISTICS OF INDIVIDUALS**								
A	Knowledge and Beliefs about the Intervention	−1*	+1	+1	+1	+1	+1	
B	Self-efficacy (First prescription)	+1	+1	+1	+2	+1	+1	
	(Deprescription)	−1	−1	−1	+1	+1*	+1	
C	Individual Stage of Change	−1	−1	ND	−1*	+2	+2	**
D	Individual Identification with Organization	ND	ND	ND	ND	ND	ND	
E	Other Personal Attributes	+1	ND	ND	+1	+2	ND	
**V. PROCESS**								
A	Planning	ND	ND	ND	ND	ND	ND	
B	Engaging	ND	ND	ND	ND	ND	ND	
1	Opinion Leaders	ND	ND	ND	ND	ND	ND	
2	Formally Appointed Internal Implementation Leaders	+1	0	0	ND	+2	+1*	*
3	Champions	ND	ND	ND	ND	ND	ND	
4	External Change Agents	ND	ND	ND	ND	ND	ND	
5	Key Stakeholders	+1	ND	−1	+1	+2	+2	**
6	Innovation Participants	−1*	ND	−1	−1	ND	+2	
C	Executing	+1	ND	ND	ND	+2	ND	

** Strongly discriminatory construct. * Weakly discriminatory construct. 0: Neutral Rating. X: Mixed Rating. +1*: mixed rating with aggregate of mixed comments positive. −1*: mixed rating with aggregate of mixed comments negative. ND: No Data. FG: Focus Group.

**Table 3 ijerph-18-07964-t003:** Participants’ supporting statements.

CFIR Constructs	HIGPs Statements	LIGPs Statements	Nº of Statements
Complexity	“… I think that the intervention was well thought out, I haven’t found it difficult at all……”	“… you get to a point when you can’t, you don’t get out the paperwork (intervention instructions) every day, because our day to day goes so fast that you have to plan it, you’re not going to open the drawer every day to see what the intervention was about.”	14
Adaptability	“…… tell them that they had to reduce their dose, we’ll reduce it by a quarter or you would tell them to file down the tablet every week with a nail file, the first week twice, bam bam, the next week, 3 or 4 times, because that will help us to stop it sooner, filing it down with a nail file because it’s very difficult to remove a quarter. They are very old and their eyesight isn’t good, the nail file works well for me…”	“I’ll have the dose and you come back to see me in a month, I’ll halve the morning dose and you come back in… or in two months, …”	25
External Policy and Incentives	“I think that the indicator is good because it’s also an indicator of poor practice, that’s why it’s there, to help you get information about how you’re doing, an indicator can also help you meet the indicator at a particular time.” “The pharmacy also gave us some leaflets to hand out about benzodiazepines……, and that also helps, you give it to the patient……”	“…that not all the health system has this culture of evaluation, in specialist care, there is no control over prescriptions, unlike us, it doesn’t matter to them if they prescribe one thing or another because nobody is going to check.”	32
Implementation Climate	“We take everything on board, we’re pioneers, everything, we sign up for everything at our center……” “In our center….it has been very well received, … what’s more, our coordinator is very interested in us doing new things, in participating in things like this, …”	“Bad in my center, anything new is a struggle”	9
Compatibility	“First I thought it was going to be a little difficult because of the stress of everyday work, this requires more time than usual…… so…… I was gradually reducing doses, then we had a period with more winter illnesses and the practice was much busier than usual etc., and this was when I stopped doing as much……”	“…but when it comes to putting it into practice, in doing it in everyday practice when you’re under pressure, when you’re up to your neck like always, it’s very difficult…”	35
Relative priority	“Rather than priority, what I liked most about it was that it raised awareness but I would put it at the same level as so many of the other interventions that we implement, …… but it is true that it has been a wake-up call, it’s raised awareness about an important is-sue. “	“……we’ve tried to prioritize…… but I don’t think they have prioritized this intervention over other things.”	18
Self-efficacy	“ This is what it has been useful for me personally, for new treatments, …trying to prevent patients from becoming chronic users, and yes it’s given me a tool that I can use to help chronic users come off the drug, a difficult task, but it’s given me a system for doing this, little by little, let’s see if we manage to help them, I’ve definitely changed my attitude towards new patients, it’s really helped me with this”	“Now when you prescribe you explain that this is a medication, for a short time, for problems… For chronic users, I often don’t even think about it, I leave things as they are.”	59
Individual Stage of Change	“It sometimes depends on your caseload, and as it’s normally high, but I think that ……, like I’ve mentioned before, in the height of winter with so many people who are sick there are times… (you stop implementing) and this is reflected in the numbers of course, but, it’s something to keep in mind and regardless or not of whether the study finishes, it’s something that has sunk in and it’s something you get used to doing…”	”…when you do the training at the beginning, you tell everyone, then you go on holidays for a month, the summer comes, and then you forget.”	22
Formally Appointed Implementation Leader Engagement	“The person who came to sell us the project has been very important in our center, …he makes everything easier… and then it makes you become more involved, you’re going to try this, you’re going to get better results…”	“Yes, I think they are very capable people, despite that, … I used it (implemented it) but I would like to have used it more…”	20
Stakeholders Engagement	“I’ve realized that it doesn’t take a lot of effort, …, just remember, make a little effort, here I’m going to rank number one in terms of users and I’ve realized that I’ve brought my numbers down simply by making a little effort, …, so yes, you try to prescribe less, ask why they are taking it, try to negotiate with your patient, reduce it a little, it’s not that you have to put up a big fight…”	“I’ve tried to follow it but sometimes you don’t do everything, just a part…, you give them information and aim to continue on other days, and that day then…it often stops there…”	35

## Data Availability

Data sharing is not applicable to this article as no datasets were generated or analyzed during the current study.

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
