# Peer review of "Evaluating the Implementation of a Multicomponent Intervention Consisting of Education and Feedback on Reducing Benzodiazepine Prescriptions by General Practitioners: BENZORED Hybrid Type I Cluster Randomized Controlled Trial"

_ijerph, 2021, doi:10.3390/ijerph18157964_

Round 1

Reviewer 1 Report

Overall this is an informative study about focusing on a qualitative evaluation of the BENZORED IV trial. 

Introduction: The introduction is clear, however needs a stronger focus on the why the implementation of changes to clinical practice have not been successful and what gaps in the literature this study will fill. Clear aims of the study also need to be described. 

Methods: has this study been reported according to the COREQ or similar qualitative reporting guideline. If so, please mention, if not, please ensure all aspects are included in the manuscript.

cfirwiki.org- this link did not work

Results: Table 1: average number of scripts would also be an interesting addition to the table.

Discussion: overall an informative discussion. Some minor grammatical errors are present

Reviewer 2 Report

The paper is  interesting so it could be published.  However should be improved.

BZD use among GP is a major topic and how they are trained on prescribed or desprescribed is relevant.

Abstract

It is clear

Introduction

There some papers that could help you to explain the topic,  background,  the lack of information, training  and the relevance of benzodiazepine misuse or abuse  among different populations.

Ribeiro PRS, Schlindwein AD. Benzodiazepine deprescription strategies in chronic users: a systematic review. Fam Pract. 2021 Apr 28:cmab017. doi: 10.1093/fampra/cmab017. Epub ahead of print. PMID: 33907803.

Richards GC, Mahtani KR, Muthee TB, DeVito NJ, Koshiaris C, Aronson JK, Goldacre B, Heneghan CJ. Factors associated with the prescribing of high-dose opioids in primary care: a systematic review and meta-analysis. BMC Med. 2020 Mar 30;18(1):68. doi: 10.1186/s12916-020-01528-7. PMID: 32223746; PMCID: PMC7104520.

de la Iglesia-Larrad JI, Barral C, Casado-Espada NM, de Alarcón R, Maciá-Casas A, Vicente Hernandez B, Roncero C. Benzodiazepine abuse, misuse, dependence, and withdrawal among schizophrenic patients: A review of the literature. Psychiatry Res. 2020 Feb;284:112660. doi: 10.1016/j.psychres.2019.112660. Epub 2019 Oct 28. PMID: 31757643.

Liu L, Jia L, Jian P, Zhou Y, Zhou J, Wu F, Tang Y. The Effects of Benzodiazepine Use and Abuse on Cognition in the Elders: A Systematic Review and Meta-Analysis of Comparative Studies. Front Psychiatry. 2020 Sep 17;11:00755. doi: 10.3389/fpsyt.2020.00755. PMID: 33093832; PMCID: PMC7527532.

Blanco C, Han B, Jones CM, Johnson K, Compton WM. Prevalence and Correlates of Benzodiazepine Use, Misuse, and Use Disorders Among Adults in the United States. J Clin Psychiatry. 2018 Oct 16;79(6):18m12174. doi: 10.4088/JCP.18m12174. PMID: 30403446.

Material and Methods

It is  difficult to read the paper with all the expressions and concerns of the participants. It could be clear if you try to summarized it in one or two tables, that allow read the paper and refer to the table if you  want to read the sentences of the doctors

Results

Table 1

It surprise me as more age as less years in the health center, could you explain this fact.  It is suppose as older you are as more years you have been working ( in the same place?)

Tale 2 underline the meaning of GD

Discussion

Should be expanded. Use the papers that I added in the introduction. Discuss the relevance of training deeply.

The  10 constructs of the 5 domains should be summarized in the first sentences. After that you should discus

Any comments on the relevance of the domains?

Can you explain/ discuss the results of table 1

How cold be improve the training in order  to reduced the perceived complexity of discontinuation of BZD,

Could you speculate how it can be linked  with discontinuation o reduction in special populations older, mental health patients, etc? any kind of special training?

Add a sentence on limitations and strengths of this study

Conclusion

I agree that is clear that medical education is critical. But I would like that the “key elements” were summarized in this final sentence.

References

Update  it.  I suggested some recent papers, see the introduction.

Reviewer 3 Report

The study was well carried out. The results were well interpreted in the Results section and well described in the Discussion. The literature is selected correctly and cited correctly in both the Introduction and the Discussion section. I believe that it will be a very helpful tool in the GPs work.

Author Response

Thank you very much for reviewing our manuscript